# Synergistic Interaction between Job Stressors and Psychological Distress during the COVID-19 Pandemic: A Cross-Sectional Study

**DOI:** 10.3390/ijerph192113991

**Published:** 2022-10-27

**Authors:** Risto Nikunlaakso, Kaisa Reuna, Kirsikka Selander, Tuula Oksanen, Jaana Laitinen

**Affiliations:** 1Finnish Institute of Occupational Health, 00032 Työterveyslaitos, Finland; 2Institute of Public Health and Clinical Nutrition, University of Eastern Finland, 70210 Kuopio, Finland

**Keywords:** work stress, mental health, interaction, social capital, COVID-19

## Abstract

Psychosocial job stressors increase the risk of mental health problems for the workers in health and social services (HSS). Although previous studies suggest that the accumulation of two or more stressors is detrimental to mental health, few studies have examined the synergistic interaction of accumulating job stressors. We examined survey responses from 9855 Finnish HSS workers in a cross-sectional study design from 2021. We conducted an interaction analysis of high job demands, low rewards and low workplace social capital on psychological distress, focusing on the relative excess risk due to interaction (RERI). Additionally, we analysed the interaction of job demands, low rewards and COVID-19 burden (extra workload and emotional load). Our analysis showed that the total RERI for the job stressors on psychological distress was considerable (6.27, 95% CI 3.14, 9.39). The total excess risk was caused by two-way interactions, especially between high demands and low rewards and by the three-way interaction of all stressors. The total RERI for job demands, low reward and COVID-19 burden (3.93, 95% CI 1.15, 6.72), however, was caused entirely by two-way interaction between high demands and low rewards. Mental health interventions tackling high demands, low rewards and low social capital are jointly needed.

## 1. Introduction

Psychosocial job stressors are a major risk for sickness absence and premature disability retirement due to mental disorders [1]. Work-related stress and sickness absences due to stress are prevalent among health and social service (HSS) workers, and the situation has worsened since the COVID-19 pandemic began [2,3]. During the pandemic, HSS workers have also faced considerable physical strain in their work, including fatigue, sleep problems, and COVID-19 infection, especially due to shift work [4,5,6]. High job demands and low rewards are among the key job stressors increasing the risk of job stress and more adverse health issues [7,8]. High workplace social capital is a potential protective factor for mental health; however, low workplace social capital is associated with self-reported depression and can thus increase worker strain [9].

Job stressors are comprehensively studied in epidemiological research; however, most previous studies primarily investigated one stress factor at a time [10]. In the everyday work environment, many stress factors, such as high workload, time pressure, violence, and harassment, coexist [11]. The situation calls for studies which investigate accumulating and interacting job stressors. Juvani and her colleagues [12] investigated how the clustering of several psychosocial stressors affects mental health among public sector workers. The study discovered that job strain, effort–reward imbalance, and organizational injustice cluster in the same individuals. Individuals who experience two or three stressors simultaneously had the highest risk of disability pension due to depression. Milner et al. [13] studied the association of several psychosocial job qualities and reporting attendance at a mental health ward, finding in a fixed effects model no relationship between psychosocial job stressors and service use.

Essentially, Juvani et al. and Milner et al. studied the effect of cumulative adversities on mental health [14]. The syndemics literature, introduced by Singer [15], focuses instead on the synergistic interaction of adversities. In a syndemic situation, two or more epidemics interact synergistically, causing excess burden on the population [16]. In the work environment, a syndemic effect could produce a vicious circle, i.e., a super-additive interaction of job stressors, which affects HSS workers’ mental health more than would be expected from the combination of coexisting job stressors. Revealing possible synergistic interactions between several stressors can facilitate more efficient interventions to promote mental health: if a synergistic interaction exists between exposures, a joint intervention addressing all the interacting exposures simultaneously is more efficient than interventions tackling single stressors [14].

Studying interaction as a departure from multiplicativity has been more common in epidemiological research [17] than studying interaction as a departure from additivity. The latter is, however, more important when studying public health: it indicates which subgroups of the study population are vulnerable to interacting exposures, and thus which subgroup would benefit most from an intervention or treatment [18]. Studying additive interaction is recommended also in the syndemics literature, being more relevant to the theory of synergistically interacting epidemics [19]. A few job stressor studies have analysed interaction as a departure from additivity [20,21,22,23,24]. Choi et al. [20], for example, discovered that workers with low job control and low social support at work were at excessive risk of psychological distress when job demands were high. These studies analysed, however, effect modification or interaction between two variables at a time. To our knowledge, no job stressor studies have been conducted on the synergistic interaction of more than two job stressors. In this paper we aim to fill this gap by expanding the analysis into a three-way interaction.

In addition to the more common job stressors, we include in our analysis the extra burden that the COVID-19 pandemic has placed on HSS workers. For example, in a Finnish cohort study, potentially traumatic COVID-19 pandemic-related events and front-line COVID-19 work were associated with personnel’s psychological distress [2]. Ervasti et al. [25] also studied the impacts of COVID-19 on psychosocial stressors; however, to our knowledge, no previous study has included the excess burden of the COVID-19 pandemic in an analysis of interacting work stressors.

The present study aimed to discover whether synergistic interaction between several job stressors occurs among HSS workers. We first examined the interaction of high job demands, low rewards, and low workplace social capital. Focusing on the relative excess risk due to interaction (RERI), we analyzed whether this accumulation is associated with workers’ psychological distress (PD). Second, we replaced social capital with COVID-19 burden—extra workload and fear for one’s own health—in the interaction model and investigated whether it affects PD. Thus, we studied the following hypotheses:

**Hypothesis** **1** **(H1).**
*High job demands, low rewards and low workplace social capital interact and cause excess risk for PD in HSS workers.*


**Hypothesis** **2** **(H2).**
*High job demands, low rewards and COVID-19 burden interact and cause excess risk for PD for HSS workers.*


## 2. Methods

### 2.1. Study Design and Population

The present study examines data collected from a survey of Finnish HSS employees. The survey has been conducted annually since 2018 and measures the psychosocial strain factors and job resources in HSS work. The survey used in this cross-sectional study was undertaken 26.10.2021–28.11.2021 in six Finnish public health and social care organizations. 11,925 employees who were actively working in the organization during that time responded to the survey. Workers on parental, sick or study leave were excluded from the eligible population. The response rate was 62%. 90% gave their consent to use the data for research. After excluding employees without research consent, with missing values on any job stressor or covariate, the final data comprised 9855 respondents.

The study was approved by the ethical board of the Finnish Institute of Occupational Health. Participation in the survey was voluntary and consent to use the responses for scientific research was requested in the questionnaire.

### 2.2. Variables

#### 2.2.1. Outcome

The main outcome, psychological distress (PD), was measured using a 4-item Patient Health Questionnaire (PHQ-4). We used it because it is brief and thus less burdensome for respondents, and yet is a validated screening instrument for psychological distress [26]. PHQ-4 comprises 2 items from the 9-item Patient Health Questionnaire (PHQ-9) and 7-item Generalized Anxiety Disorder (GAD-7). Responses were scored as 0 (“not at all”), 1 (“several days”), 2 (“more than half the days”), or 3 (“nearly every day”). The total score thus ranged from 0 to 12. The internal reliability of the scale was good, as the average inter-correlation among the items was high (Cronbach’s alpha = 0.86). Following the rationale of Löwe et al. [27], individual survey responses with a PHQ-4 score of 6 or more were coded as psychologically distressed.

#### 2.2.2. Exposure Variables

To measure psychosocial job stressors—job demands and job rewards—we chose the commonly used and validated instruments from Karasek [28] and Siegrist [29]. In our survey, to reduce response burden, job demands were measured with two items (Cronbach’s alpha = 0.88) derived from the Job Content Questionnaire [28]: “An unreasonable amount of work is expected of me” and “I don’t have enough time to get my work done”. The response scale was five-level (1 = strongly agree to 5 = strongly disagree). We calculated a mean of the scores. Job rewards were measured with three items (Cronbach’s alpha = 0.74) from the effort–reward imbalance model [29]: “How much do you feel you get in return for work in terms of income and job benefits?”, “How much do you feel you get in return for work in terms of recognition and prestige?”, and “How much do you feel you get in return for work in terms of personal satisfaction?”. The response scale was five-level (1 = very much to 5 = not at all) and we calculated a mean of the scores. Workplace Social capital was measured using a validated and psychometrically tested self-assessment scale [9]. We calculated a mean from eight items (Cronbach’s alpha = 0.87): “We have a ‘we are together’ attitude”, “People feel understood and accepted by each other”, “We can trust our supervisor”, “People in the work unit cooperate in order to help develop and apply new ideas”, “Our supervisor treats us with kindness and consideration”, “Our supervisor shows concern for our rights as an employee”, “People keep each other informed about work-related issues in the work unit” and “Do members of the work unit build on each other’s ideas in order to achieve the best possible outcome?”. The response scale had five points (1 = strongly disagree to 5 = strongly agree in the first seven items and 1 = to a very little extent to 5 = to a very great extent in the last item). To examine interaction effects in strained workers, we set the lowest quintile of job demands and of social capital, and the highest quintile of rewards to indicate exposure to each job stressor. The remaining four quintiles of each stressor were set as non-exposed.

COVID-19 burden was measured using two statements created by the authors in collaboration with the studied health and social care organizations: “I have been afraid for my health because of the COVID-19 situation”, and “my workload has increased because of the COVID-19 situation”, with a response scale of yes/no. Respondents who answered yes to both statements were classified as having COVID-19 burden.

### 2.3. Covariates

Employee age, gender, supervisory position (yes/no), and occupation were collected from organizational registers. We classified the respondents into three levels of socio-economic status according to their occupation: Skilled labor included, for example, managers, physiotherapists, physicians, and nurses. Semi-skilled labor included, for example, practical nurses, social workers, and clerical workers. Unskilled labor included, for example, cleaning and kitchen staff. We also included self-reported health status as a possible confounding factor. It was measured with a question: “how is your health”, with a 5-point scale: good, fairly good, average, fairly poor, and poor. We used this 5-point classification in our analyses.

### 2.4. Statistical Analysis

During statistical analysis, we calculated and compared relative excess risks due to the interaction (RERI) of the job stressors. First, using binary logistic regression, we calculated odds ratios (OR) for PD in relation to exposure variables alone and to their interaction terms. As advised by Knol et al. [30], we calculated the OR:s for the interaction terms by multiplying their OR:s by the interacting exposure variables. This approach works well when the outcome is rare, as RERI_OR_ approximates the relative excess risk due to the interaction of risk ratios, RERI_RR_. For example, in a case of two interacting exposure variables A and B,
ln(odds) = β_0_ + β_1_A + β_2_B + β_3_AB
the combined effect of A and B, compared with no effect of A and B, is [30]:OR_A+B+_ = e^β1+β2+β3^ = e^β1^ × e^β2^ × e^β3^ = OR_A_ × OR_B_ × OR_AB_

In a case of two interacting exposure variables A and B, relative excess risks due to interaction, RERI_OR_, are calculated as follows [18]:RERI_OR2_ = OR_A+B+_ − OR_A+B−_ − OR_A-B+_ + 1 = e^β1+β2+β3^ − e^β1^ − e^β2^ +1

With the analysis of three interacting variables, we followed the instructions by Katsoulis et al. [31]. In a case of three interacting exposure variables A, B, and C,
ln(odds) = β_0_ + β_1_A + β_2_B + β_3_C + β_4_AB + β_5_AC + β_6_BC + β_7_ABC

The total relative excess risk due to interaction (TotRERI_RR3_, or TotRERI_OR3_, when the outcome is rare) should be calculated by comparing the joint effect of the three risk factors to the situation when each one acts separately:TotRERI_OR3_ = OR_A+B+C+_ − OR_A+B−C−_ − OR_A−B+C−_ − OR_A−B−C+_ + 2 = e^β1+β2+β3+β4+β5+β6+β7^ − e^β1^ − e^β2^ − e^β3^ + 2

As the total RERI can be a result of either three- or two-way interactions, we need to calculate the effect of three-way interaction RERI_OR3_ as follows:RERI_OR3_ = OR_A+B+C+_ − OR_A+B+C__−_ − OR_A+B__−C+_ − OR_A__−B+C+_ + OR_A+B-C__−_ + OR_A__−B+C−_ + OR_A__−B−C+_ − OR_A__−B−C−_ = e^β1+β2+β3+β4+β5+β6+β7^ − e^β1+β2+β4^ − e^β1+β3+β5^ − e^β2+β3+β6^ + e^β1^ + e^β2^ + e^β3^ − 1

The interaction of the three factors is positive or super-additive when TotRERI_OR3_, RERI_OR3_, or RERI_OR2_ is above zero. Correspondingly, the interaction is negative or sub-additive when TotRERI_OR3_, RERI_OR3_, or RERI_OR2_ is below zero [18].

Using the formula suggested by Katsoulis et al. [31], we created two interaction models. Model 1 investigates the interaction of high job demands, low rewards, and low social capital. We calculated first, using binary logistic regression, the main effects, and then multiplicative interaction effects of the stressors. We then calculated total RERI, two-way RERI:s, which we stratified by the third factor, and three-way RERI. In model 2, we replaced low social capital with COVID-19 burden. Both models are adjusted for covariates. For the estimation of 95% confidence intervals, we used the delta method.

## 3. Results

Table 1 shows the characteristics of the respondents in the study population. Respondents were mostly women and had a socio-economic status of skilled labor. Most respondents also reported having good health. One tenth of the individual respondents reported intermediate or severe PD. 19.4% of the respondents reported having extra burden because of the COVID-19 situation.

Table 2 shows the model 1 with OR:s of job stressors and of their interactions and the RERI:s for two- and three-way interactions on PD. Adjusting for age, gender, supervisory position, occupation and self-reported health status, all job stressors had alone a strong association with PD. In accordance with hypothesis 1, The total RERI was 6.27 (95% CI 3.14–9.39), suggesting a six-fold risk due to the joint presence of all stressors, compared to each stressor acting separately. The excess risk was caused by both two- and three-way interactions. The interaction between high demands and low rewards, whether in the stratum of low social capital (RERI 3.19, 95% CI 1.55–4.83) or intermediate or high social capital (RERI 3.26, 95% CI 1.13–5.39), produced a high RERI for PD. The interaction between low rewards and low social capital in the stratum of high demands was also associated with PD. Contrarily, the interaction between high demands and low social capital was significant when low rewards was absent (−1.37, 95% CI −2.69, −0.04). This indicates a sub-additive interaction, meaning that the risk of PD is lower when high demands and low social capital interact. Finally, the RERI for three-way interaction of the risk factors was 3.41 (95% CI −0.53, 7.36), indicating a strong joint effect also without the contribution of two-way interactions. The RERI for three-way interaction had, however, a relatively high *p*-value, 0.090, indicating relatively weak evidence of the effect.

In model 2 (Table 3), we replaced low social capital with COVID-19 burden to see whether high demands, low rewards and COVID-19 burden interact and increase the risk of PD. As in model 1, high demands and low rewards were alone strongly associated with PD, while COVID-19 burden was less so. The total RERI of the three stressors was 3.93 (95% CI 1.15, 6.72), suggesting a nearly four-fold risk due to joint presence of all stressors, compared to each stressor acting separately. However, contrary to model 1 and hypothesis 2, the excess risk was entirely caused by two-way interaction between high demands and low rewards.

## 4. Discussion

The present study examined the synergistic interaction between job stressors, finding out whether interaction is associated with psychological distress (PD) among health and social service (HSS) workers. Focusing on the additive interaction effect, we found support for our first hypothesis: the relative excess risk due to interaction (RERI) for high job demands, low rewards and low social capital was noticeable. Having the three stressors simultaneously raised the risk of having PD over six times compared to the stressors acting separately. The excess risk was caused by both two- and three-way interactions, with high job demands and low rewards producing a significant RERI regardless of workplace social capital. Adding low social capital to the interaction increased the risk: the three-way interaction produced a RERI of 3.41 (95% CI −0.53,7.36), although the evidence on the three-way interaction was rather weak (*p*-value 0.090), possibly due to limited data size.

To our knowledge, this is the first study that has examined the synergistic interaction effect of three job stressors on workers’ health. Our results are in line with those of Juvani et al. [12], who found that the accumulation of several job stressors is detrimental to mental health, and Selander et al. [32], who discovered that the accumulation of several job stressors is associated with low work ability. The results of this study are also somewhat comparable to those of Choi et al. [20], who found a strong synergistic effect between job control and social support at work when job demands were low.

In the analysis of the interaction between high demands, low rewards and COVID-19 burden, we found that job demands and low rewards accounted for the total excess risk for PD. Refuting hypothesis 2 of this study, COVID-19 burden had no effect on the excess risk. The result indicates that HSS workers are under such a severe strain that COVID-19 burden poses no extra burden on them. Another possible explanation is our study measure: questions about fear for one’s own health and extra workload may not have captured the essence of COVID-19 burden.

Evident limitations must be considered when generalising the results of this study. First, our data size was relatively small for examining three-way interactions. This is a recognised challenge in interaction analyses [17]. More voluminous data would have enabled more reliable evidence on interaction effects, especially on the three-way interaction effect of high demands, low rewards, and low social capital on PD. The findings of this study should be confirmed with more data in the future. Second, the current study used cross-sectional data. We were able to analyse only the OR:s of interacting exposures and the prevalence of PD. Even though the outcome in our analysis was rare, OR:s only approximate risk ratios. Analysing the risk ratios of exposures and the incidence of PD in a longitudinal design would produce more reliable evidence of the synergistic interaction of several stressors. The unexpected sub-additive interaction effect (RERI −1.37, 95% CI −2.69, −0.04) between high demands and low social capital in the stratum of intermediate or high rewards should be examined with longitudinal analysis, for example. Third, both the exposure variables and the outcome in our study, PD, was measured with self-reported measurements, which may increase subjectivity and reporting bias [12]. However, as PD refers to symptoms of mental disorders, it identifies individuals who are under considerable stress but who can still be helped to avoid work disability due to mental disorders. Future studies are needed to investigate the effect of synergistic interaction on more severe outcomes, such as sickness absences.

Despite these limitations, this study has important implications. The results of the present study suggest that the synergistic interaction of several job stressors may increase the risk of having psychological distress, which can ultimately lead to sickness absences, more adverse mental disorders, and even work disability pension. Our results show, however, that not all stressors create a synergistic interaction. Future studies should, thus, examine which job stressors are most important in creating synergistic interaction and increasing the risk for PD. This information is needed to identify and monitor the most important psychosocial risk factors at HSS workplaces.

## 5. Conclusions

The results of this study emphasize the need for HSS workplaces to identify job stressors, for example, with annual well-being surveys and to support the health and work ability of individuals who are under risk of accumulating job stressors. Interventions that decrease the excess risk of interacting job stressors are also needed. Following the argument of Tsai and Venkataramani [14], the findings of this study suggest that interventions tackling high demands, low rewards, and low social capital could decrease PD in isolation. They would, however, have a greater preventive impact if the intervention would address all stressors jointly. Organizational workplace interventions which tackle job stressors and promote better workplace social capital are seldom studied; instead, intervention studies usually focus on the individual level, for example, through the use of stress coping skills and mental relaxation [33]. Organizational-level interventions are urgently needed in the future to maintain HSS workers’ mental health and work ability, and to attract new personnel into HSS.

## Figures and Tables

**Table 1 ijerph-19-13991-t001:** Characteristics of the study population (N = 9855).

	%
Sex	
Female	88.8
Male	11.2
Age	
Under 40	34.5
40–49	24.9
50–59	28.0
Over 59	12.6
Supervisors	5.8
Socio-economic status	
Skilled labour	62.2
Semi-skilled labour	34.1
Unskilled labour	3.7
Self-reported health good	74.8
Experiencing COVID-19 burden	19.4
Psychological distress	10.4

**Table 2 ijerph-19-13991-t002:** Odds ratios of high demands, low rewards, and low social capital and their production terms from a binary logistic regression analysis of psychological distress on an individual level, with indices of additive (relative excess risk due to interaction) and multiplicative interaction.

Risk Factors	N with PD/Total N	OR	95% CI	*p*-Value
high demands only	132/870	3.17	2.51, 4.00	0.000
low rewards only	117/830	2.37	1.86, 3.02	0.000
low social capital only	85/813	2.09	1.60, 2.73	0.000
**Measure of interaction: multiplicative**				
high demands × low rewards	113/330	1.04	0.70, 1.54	0.859
high demands × low social capital	41/273	0.44	0.27, 0.71	0.001
low rewards × low social capital	123/515	0.89	0.59, 1.33	0.579
high demands × low rewards × low social capital	140/324	1.87	0.96, 3.65	0.066
**Measure of additive interaction: relative excess risk due to interaction (RERI)**				
	**RERI**	**95% CI for RERI**		***p*-value**
RERI_2_ (high demands × low rewards/intermediate or high social capital)	3.26	1.13, 5.39		0.003
RERI_2_ (high demands × low social capital/intermediate or high rewards)	−1.37	−2.69, −0.04		0.043
RERI_2_ (low rewards × low social capital/low or intermediate demands)	0.97	−0.23, 2.16		0.115
RERI_2_ (high demands × low rewards/low social capital)	3.19	1.55, 4.83		0.000
RERI_2_ (high demands × low social capital/low rewards)	0.86	−0.68, 2.40		0.274
RERI_2_ (low rewards × low social capital/high demands)	1.38	0.22, 2.54		0.020
RERI_3_ (high demands × low rewards × low social capital)	3.41	−0.53, 7.36		0.090
TotRERI_3_ (high demands × low rewards × low social capital)	6.27	3.14, 9.39		0.000

PD = Psychological distress; OR = Odds ratio; CI = Confidence interval; RERI = Relative excess risk due to interaction; OR:s are adjusted for age, gender, supervisory position, occupation and self-reported health status.

**Table 3 ijerph-19-13991-t003:** Odds ratios of high demands, low rewards, and COVID-19 burden and their production terms from a binary logistic regression analysis of psychological distress on an individual level, with indices of additive (relative excess risk due to interaction) interaction.

Risk Factors and Their Product Terms	N with PD/Total N	OR	95% CI	*p*-Value
high demands only	121/837	2.83	2.22, 3.60	0.000
low rewards only	186/1 035	2.97	2.40, 3.68	0.000
COVID-19 burden only	89/1 065	1.43	1.11, 1.85	0.006
**Measure of interaction: multiplicative**				
high demands × low rewards	158/423	1.09	0.75, 1.57	0.689
high demands × COVID-19 burden	52/306	0.80	0.51, 1.26	0.334
low rewards × COVID-19 burden	54/310	0.66	0.43, 1.03	0.059
high demands × low rewards × COVID-19 burden	95/231	1.32	0.67, 2.60	0.397
**Measure of additive interaction: relative excess risk due to interaction (RERI)**				
	**RERI**	**95% CI for RERI**		***p*-value**
RERI_2_ (high demands × low rewards/no COVID-19 burden)	4.34	2.15, 6.54		0.000
RERI_2_ (high demands × COVID-19 burden/intermediate or high rewards)	−0.01	−1.24, 1.21		0.983
RERI_2_ (low rewards × COVID-19 burden/low or intermediate demands)	−0.58	−1.66, 0.51		0.296
RERI_2_ (high demands × low rewards/COVID-19 burden)	3.16	1.04, 5.29		0.004
RERI_2_ (high demands × COVID-19 burden/low rewards)	0.06	−1.11, 1.22		0.924
RERI_2_ (low rewards × COVID-19 burden/high demands)	−0.14	−1.39, 1.11		0.826
RERI_3_ (high demands × low rewards × COVID-19 burden)	0.18	−3.50, 3.86		0.922
TotRERI_3_ (high demands × low rewards × COVID-19 burden)	3.93	1.15, 6.72		0.006

PD = Psychological distress; OR = Odds ratio; CI = Confidence interval; RERI = Relative excess risk due to interaction; OR:s are adjusted for age, gender, supervisory position, occupation and self-reported health status.

## Data Availability

Data are available upon reasonable request. The deidentified data and statistical analysis code that support the findings of this study are available on reasonable request from the corresponding author. The data are not publicly available due to legislative restrictions, as the data contains information that could compromise the privacy of the research participants.

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
