# Peer review of "Synergistic Interaction between Job Stressors and Psychological Distress during the COVID-19 Pandemic: A Cross-Sectional Study"

_ijerph, 2022, doi:10.3390/ijerph192113991_

Round 1
Reviewer 1 Report
Maybe it is worth more to add more information about the respondents, because they are too general. Overall a valuable article, congratulations.
Author Response
Thank you for the comments and the opportunity to respond to them. We very much appreciate the points and have now revised the manuscript accordingly:
- We added more information about the respondents’ occupation and socio-economic status. We now write in the Covariates section, on page 4, within lines 152–156: “We classified the respondents into three levels of socio-economic status according to their occupation: Skilled labor included for example managers, physiotherapists, physicians, and nurses. Semi-skilled labor included for example practical nurses, social workers, and clerical workers. Unskilled labor included for example cleaning and kitchen staff.”
- We had already a table of the characteristics of the respondents in the beginning of results section on page 5. In addition to the table, we have written within lines 210–212 as follows: “Respondents were mostly women and had a socio-economic status of skilled labor. Most respondents also reported having good health.”
Reviewer 2 Report
This paper conducted a large scale survey among Finnish HSS workers for an interaction analysis of high job demands, low rewards and low workplace social capital (or COVID-19 burden) on psychological distress. It addresses an important knowledge gap that is, different job stressors may interact and generate synergistic effects on certain work population. The results showed significant results where multiple stressors -- high job demands, low rewards and low social capital, increased PD by six times. This finding is of important value for future intervention studies. The paper is generally well written. Nevertheless, justification for the choices of survey questions appears to be missing. For a survey of this scale, it is important to justify how the outcome and exposure measures are chosen. For example, whether the questionnaire is developed by preliminary research or based in literature review. In “Variables” section, the authors explained how items were selected from established measurement scales. They just need to add one or two paragraphs here to explain why such scales and items were selected. In line 183, there is a typo. It should be “…by Katsoulis…”.
Author Response
Thank you for the comments and the opportunity to respond to them. We very much appreciate the suggestions and have now revised the manuscript accordingly:
- On page 3, in “Study design and population” section, within lines 97–98, we added the following text: “The survey has been conducted annually since 2018 and it measures the psychosocial strain factors and job resources in HSS work. The survey used in this cross-sectional study…”
- In “Variables” section, we made the following additions:
- within lines 111–112 we now write: “We used it because it is brief and thus less burdensome for respondents, and yet a validated screening instrument for psychological distress[26].”
- within lines 120–121 we now write: “To measure psychosocial job stressors—job demands and job rewards—we chose the commonly used and validated instruments from Karasek [28] and Siegrist [29].”
- within lines 121–123 we now write: “In our survey, to reduce response burden, job demands were measured with two items derived from the Job Content Questionnaire[28]:…”
- within lines 131–132 we now write: “Workplace Social capital was measured using a validated and psychometrically tested self-assessment scale [9]”.
- We also calculated and added internal variability measures (Cronbach’s alpha) for PHQ-4 (lines 116–117), job demands (line 126), low rewards (line 131), and workplace social capital (line 133).
- Thank you also for mentioning the typo in line 183 (or 202 in the revised manuscript). We have now corrected it.
Reviewer 3 Report
In Method sections, the sample of HSS workers should also include some sample job titles like counselors, social workers, case managers.
I am wondering whether in addition to psychological stressors, there might be physical and physiological stressors which might also be specifically identified. COVID-19 had people working long hours, double-shifts, sleeping in facilities that needed to be in-person. These are not psychological stressors per se.
Author Response
Thank you for the comments and the opportunity to respond to them. We very much appreciate the suggestions and have now revised the manuscript accordingly:
- We now write on page 4, within lines 152–156: “We classified the respondents into three levels of socio-economic status according to their occupation: Skilled labor included for example managers, physiotherapists, physicians, and nurses. Semi-skilled labor included for example practical nurses, social workers, and clerical workers. Unskilled labor included for example cleaning and kitchen staff.”
- We also added text and citations in the introduction, on page 1, about physical strain that COVID-19 has caused. We now write within lines 29–31: “During the pandemic, HSS workers have also faced considerate physical strain in their work, including fatigue, sleep problems, and COVID-19 infection especially due to shift working [4-6].”
Reviewer 4 Report
First of all, congratulation for your work.
This study has a strong potential, but the way it is written now has some limitations.
The study has an excellent sample, 9855 participants. But the weakness starts in the introduction. The analysis is quantitative and transversal and seems to follow a hypothetic-deductive logic, but the paper does not present the empirical hypothesis and your theoric fundamentations.
The measures used in this study aren´t well presented and described, and we don´t know the reliability values of the instruments used.
The analysis is limited, and we don´t have access to much statistical data. The study is impossible to replicate.
Author Response
Thank you for the comments and the opportunity to respond to them. We very much appreciate the suggestions and have now revised the manuscript accordingly. Our responses to the specific comments are as follows:
- We have stated the two hypotheses of this study in the end of introduction. We now state on page 2, within lines 86–92 the following: “Thus, we studied the following hypotheses:
H1. High job demands, low rewards and low workplace social capital interact and cause excess risk for PD for HSS workers.
H2. High job demands, low rewards and COVID-19 burden interact and cause excess risk for PD for HSS workers.”
We also write now in the results section, on page 5, in line 220: “In accordance with hypothesis 1,…”, and on page 6, in line 243: “However, contrarily to model 1 and hypothesis 2,…”. Furthermore, we have added the following texts in the discussion: on page 7, within lines 252–253: “Focusing on the additive interaction effect, we found support for our first hypothesis:…” and on page 8, in line 271: “Refuting hypothesis 2 of this study, COVID-19 burden…”
We consider that the theoric fundamentation of this study is the possible syndemic effect, i.e. excess risk or vicious circle, caused by the interaction of job stressors. We have tried to open the fundaments of syndemic situation and interaction of exposure variables both as a departure from additivity and multiplicativity in the introduction, in the first two paragraphs of page 2. We hope our rationale is sufficient without further text changes.
- To improve the presentation and description of study variables, we have added the following texts in the methods section:
-
- In “Variables” section, we made the following additions:
-
- within lines 111–112 we now write: “We used it because it is brief and thus less burdensome for respondents, and yet a validated screening instrument for psychological distress[26].”
- within lines 120–121 we now write: “To measure psychosocial job stressors—job demands and job rewards—we chose the commonly used and validated instruments from Karasek [28] and Siegrist [29].”
- within lines 121–123 we now write: “In our survey, to reduce response burden, job demands were measured with two items derived from the Job Content Questionnaire[28]:…”
- within lines 131–132 we now write: “Workplace Social capital was measured using a validated and psychometrically tested self-assessment scale [9]”.
- We also calculated and added internal variability measures (Cronbach’s alpha) for PHQ-4 (lines 116–117), job demands (line 126), low rewards (line 131), and workplace social capital (line 133).
- The analyses of this study are focused on testing of two hypotheses, and as other referees have not commented the analyses being limited, we consider expansion of the analyses unnecessary. We have reconsidered, however, the data availability statement and to enable possible access to the data, we now state as follows: “Data are available upon reasonable request. The deidentified data and statistical analysis code that support the findings of this study are available on reasonable request from the corresponding author. The data are not publicly available due to legislative restrictions, as the data contains information that could compromise the privacy of the research participants.”
Reviewer 5 Report
Dear authors, I thank you for the opportunity to evaluate the study developed. Initially, I consider necessary and opportune studies of this nature for the field of worker's mental health.
Within reference 30: "Organizational workplace interventions that address work stressors and promoting better social capital in the workplace are quite rare" I suggest that you briefly add the reasons that make it difficult to promote these interventions to add value to the study and justify the use of the reference.
Author Response
Thank you for the comments and the opportunity to respond to them. We very much appreciate the suggestions and have now revised the manuscript accordingly:
We now write on page 9, within lines 311–316: “Organizational workplace interventions which tackle job stressors and promote better workplace social capital are seldom studied; instead, intervention studies usually focus on the individual level, for example stress coping skills and mental relaxation [30]. Organizational-level interventions are urgently needed in the future to maintain HSS workers’ mental health and work ability, and to attract new workforce into HSS.”
Round 2
Reviewer 4 Report
I thank the authors for the improvements made to the manuscript.
I believe that the reasoning behind the hypotheses could still be improved. But I recognize the improvement of the article as a whole.
In statistical terms, Cronbach's alpha is not the average of the scores but the average of the correlation between the items that are part of the scale. Correcting this note in your work when you refer to Cronbach's alpha would be significant.
Thank you very much for your work and for your attention to the comments.
Author Response
Thank you for the positive feedback. We also thank for the additional comments and the opportunity to respond to them. We have now revised the manuscript accordingly:
We clarified the meaning of internal consistency on page 3, in the Outcome section. We now write within lines 116–117: “The internal reliability of the scale was good, as the average inter-correlation among the items was high (Cronbach’s alpha = 0.86).” To indicate that Cronbach’s alpha is a measure of the correlation of the items and not the average score of the items, we also changed the place of the bracketed information regarding Cronbach’s alpha in the Exposure variables section for each three job stressors.
We could not, however, identify how we could improve the reasoning behind the study hypotheses. If the introduction still needs improvement, could it be possible that we received more specific instructions on the sentences or the parts of the text that need clarification?